# Does culture moderate the relationships between rumination and symptoms of posttraumatic stress disorder and depression?

**James Haoxiang Li**[1‡]*, **Bryan Lee**[1‡]*, **Tamsyn Reyneke**[1], **Shamsul Haque**[2], **Siti Zainab Abdullah**[2], **Britney Kerr Wen Tan**[2], **Belinda Liddell**[3], **Laura Jobson**[1]

**1** Turner Institute for Brain and Mental Health and School of Psychological Sciences, Monash University, Melbourne, Australia, **2** Department of Psychology, Jeffrey Cheah School of Medicine and Health Sciences, Monash University, Subang Jaya, Malaysia, **3** School of Psychology, University of New South Wales, Kensington, Sydney, Australia

‡ HL and BL are contributed equally to this work and are joint first authors.
* jamesli9989@gmail.com (JHL); bryanjl@student.unimelb.edu.au (BL)

## Abstract

Brooding rumination is positively associated with symptoms of both depression and posttraumatic stress disorder (PTSD). However, non-clinical cross-cultural research indicates that culture may influence these associations. This study aimed to examine the moderating effect of cultural group (Australian versus Malaysian) on the associations between brooding rumination and symptoms of depression and PTSD. European Australians ($n$ = 109) and Malaysians of varying Asian heritages ($n$ = 144) completed an online questionnaire containing the Hospital Anxiety and Depression Scale, PTSD checklist for DSM-5 and the Ruminative Response Scale-Short Form. First, Malaysian participants had higher brooding rumination than Australian participants. Second, higher levels of brooding rumination were positively associated with depression and PTSD symptom severity. Third, contrary to our expectations, cultural group did not moderate the relationships between brooding rumination and symptoms of depression and PTSD. If replicable, these results suggest that existing assessment and treatment approaches that target brooding rumination may apply to Malaysian individuals with depression and PTSD.

## Introduction

Brooding rumination, defined as repetitively dwelling on the causes, consequences and meaning of one's negative emotions, is a transdiagnostic process associated with the etiology and maintenance of several psychiatric disorders [1]. Posttraumatic stress disorder (PTSD) and depression are two psychiatric disorders that have received considerable attention in terms of brooding rumination and symptom severity. Globally, depression and PTSD are pervasive debilitating mental health disorders, recognized in most societies and cultures, and associated

**Data availability statement:** The data is available at https://osf.io/g6h8a/.

**Funding:** The authors received no specific funding for this work.

**Competing interests:** The authors have declared that no competing interests exist.

with numerous adverse outcomes for affected individuals and society [2, 3]. Considering the prevalence and adverse consequences of PTSD and depression worldwide, it is important to investigate factors that may contribute to these disorders and identify how culture may influence such factors.

Research has consistently implicated brooding rumination in the onset and maintenance of depression. For instance, brooding rumination predicts the onset of depression, serves as a moderator between negative affect and depression symptoms, and mediates the relationship between several known vulnerabilities of depression and depression symptomatology [4–6]. The Response Styles Theory [7] posits that brooding rumination triggers and intensifies depression by a) prompting individuals to think negatively about their past, present and future, b) impeding effective problem-solving, c) inhibiting the use of goal-directed behaviour, and d) eroding social support.

Brooding rumination is also consistently associated with higher PTSD symptomatology. Two recent meta-analyses found moderate to strong associations between rumination and PTSD symptoms [8, 9]. Researchers propose that that repetitively dwelling on the negative outcomes of trauma can generate more negative trauma-related appraisals, thus increasing arousal and anxiety symptoms [10–12]. Consequently, this enhances the association between traumatic stimuli and fear and perpetuates PTSD symptoms through memory biases [10–12]. Brooding rumination can also impede PTSD treatment efforts [13].

Thus, there is considerable evidence demonstrating the relationship between brooding rumination and these two psychiatric conditions, which in turn informs current psychological treatments. However, this evidence base is almost entirely derived from Western samples. Consequently, the dominant paradigm that suggests rumination is universally associated with increased psychopathology, is a notion based largely on Western research and cultural values. This is a key limitation given that (1) globally, many individuals with depression and PTSD do not come from Western cultural backgrounds, and (2) accumulating research suggests that the relationship between brooding rumination and psychopathology may vary between different cultural groups. It is important to address this concerning limitation because clinical guidelines highlight the importance of cultural tailoring in clinical practice, yet there is little research to support these practices [2, 14], and treatment effects improve significantly when interventions are culturally-tailored [15]. Thus, it is crucial to explore how culture may moderate the relationship between brooding rumination and psychopathology to improve culturally-tailored treatment approaches.

Accumulating research indicates that brooding rumination is more common in Asian than Western individuals [16]. Despite this, brooding rumination has been found to be less associated with poor psychological outcomes, including anxiety, depression and life dissatisfaction, in Asian individuals compared to Western individuals [17, 18]. Two key cultural differences have been proposed to account for these findings. First, Asian collectivistic cultures tend to value negative emotions more favourably than Western, individualistic cultures [16, 19] and often encourage brooding rumination and self-criticism as a means of maintaining group harmony [20–22]. As such, Asian individuals may be less likely to get 'stuck' in their negative emotions, leaving more room for flexible and adaptive reflection [16]. Second, in Asian collectivistic cultures, when compared to Western individualistic cultures, there is an increased tendency to self-distance from emotional experiences and focus more on contextual factors [16]. Thus, thinking repetitively about the past may be less likely to increase distress in individuals from Asian cultures. While these studies indicate that cultural factors may influence the psychological outcomes of brooding rumination, little research has examined cultural differences in brooding rumination in clinical populations and of the very few studies that have, all have been conducted within Western countries [23].

Additionally, cross-cultural non-clinical research in this area is limited by its sampling of mostly American and Chinese populations. Australia and Malaysia are two heavily under-represented populations. While Australia and America are both portrayed as individualistic societies [24], Australians have lower autonomy needs than Americans and may not be as individualistic as America [25]. Similarly, Malaysia is a multi-cultural society primarily populated by individuals of Malay, Chinese and Indian heritage [26]. Like China, Malaysia is considered to be collectivistic [26]. However, the presence of other cultural factors within Malaysian societies may influence their social emphasis on collectivism [27]. For instance, unlike Chinese cultures which base their collectivisms on Confucianism, Malaysians also obtain their emphasis on collectivism from the traditional cultures of Islam and the indigenous Malay [27]. Hence, it is crucial to explore the relationships between brooding rumination and psychological adjustment in varying Western individualistic and Asian collectivistic cultures.

This study aimed to investigate the moderating effect of cultural group on the association between brooding rumination and symptoms of PTSD and depression within an Asian collectivistic (Malaysia) and Western individualistic (Australia) cultural context. We hypothesized that there would be greater brooding rumination reported by the Malaysian group than the Australian group (Hypothesis 1). Second, brooding rumination would be positively associated with both depression and PTSD symptom severity (Hypothesis 2). Third, cultural group would moderate the associations between brooding rumination and both depression and PTSD symptoms, such that brooding rumination would be more strongly associated with symptom severity in the Australian group than Malaysian group (Hypothesis 3).

## Method

### Participants

Given the prevalence of depression and trauma-exposure in the general community, a general community sample provided a wide range of symptomatology allowing for the hypothesized relationships to be examined. Utilising G*Power [28], with a small-moderate effect size ($f^2$ = .14) [29], an alpha of .05 and power of .80, 101 participants (50 per cultural group) were found to be sufficient to power all hypotheses. Using convenience sampling, 253 participants (109 Australian, 144 Malaysian) were recruited from the general community. Eligibility criteria included being a) between 18 and 65 years of age, b) able to read and complete the questionnaire in English or Malay, and c) either residing in Australia and identifying as having European heritage or residing in Malaysia and identifying as having either Malay, Indian or Chinese heritage. Additionally, to be included in the PTSD analyses, participants had to have been exposed to a Criterion A trauma experience as indexed on the Life Events Checklist-5 [30] (Australian $n$ = 107, Malaysian $n$ = 121).

### Measures

**Hospital Anxiety and Depression Scale (HADS).** The depression subscale of the HADS (HADS-D) [31] was used to measure depression symptoms. The subscale contains seven items scored on 4-point Likert-type scales (0 = *not at all* to 3 = *definitely*). Individual item scores were summed (five items are reversed scored) to give a total depression score ranging from 0 to 21, with higher scores indicating greater depression symptom severity. This measure has demonstrated good validity and reliability, including in cross-cultural research [32]. In this study the HADS-D demonstrated acceptable internal consistency (Australia α = .78, Malaysia α = .80).

**PTSD Checklist for DSM-5 (PCL-5) with Life Events Checklist for DSM-5 (LEC-5).** The PCL-5 with LEC-5 [30] assessed trauma exposure and PTSD symptom severity.

The LEC-5 screens for exposure to traumatic events in a respondent's lifetime. Participants identify their worst single traumatic event; the *index trauma*. The LEC-5 is not scored, but used to ensure trauma exposure (i.e., eligibility criteria) and categorise participants' index traumas. The PCL-5 checklist is a 20-item self-report questionnaire that assesses PTSD symptom severity in relation to the respondent's index trauma. Items are scored using 5-point Likert scales (0 = *not at all* to 4 = *extremely*). Scores are summed to yield a total score ranging from 0–80, with higher scores indicating greater PTSD symptom severity. The PCL-5 has demonstrated good psychometric properties, including in cross-cultural research [33]. In the present study, internal consistency was excellent (Australian α = .95; Malaysian α = .96).

**Ruminative Response Scale-Short Form (RRS-SF).** Brooding rumination was measured using the brooding subscale of the RRS-SF (RRS-B) [34]. The five items were scored on 4-point Likert-type scales (1 = *almost never* to 4 = *almost always*). Item scores are summed and range from 5 to 20, with higher scores indicating greater levels of brooding rumination. The RRS-B has been shown to have good reliability and validity, including in cross-cultural research [35]. The RRS-B demonstrated good internal consistency in this study (Australian α = .79, Malaysian α = .84).

**Conscientious Responders Scale (CRS).** The CRS was utilised to identify participants who answered questions hastily and inaccurately [36]. The five items were interspersed throughout the questionnaire. Each item directly instructs participants on how to answer and is scored as either correct (1) or incorrect (0). Scores ranged from 0–5, and participants who scored less than 3 on the CRS were excluded from data analyses, based on less than 3% of random responders achieving a score of 3 or above by chance alone [36, 37].

## Procedure

Ethical approval was obtained from BLINDED. To ensure cultural appropriateness, the study was co-designed by Australian and Malaysian researchers. Following the gold standard approach [38], English measures were translated into Malay and then back-translated into English. The study was advertised on social media (Facebook and Gumtree). Interested respondents contacted the researchers via email and received a link to the online questionnaire (Qualtrics). Participants completed the PCL-5 with LEC-5, HADS-D, RRS-B and demographics. Participants were reimbursed with gift vouchers.

## Statistical analyses

All statistical analyses were conducted using SPSS (version 27.0). As some variables did not meet assumptions for normality, bootstrapping was utilised for all analyses. Given significant group differences in age, education and religion, these variables were included as covariates. A subset of the sample (*n* = 228) was included in the PTSD analyses; to be included in the PTSD analyses participants had to have been exposed to a Criterion A trauma experience. To test Hypothesis 1, a one-way analysis of covariance was conducted comparing the Malaysian and Australian groups with brooding rumination as the dependent variable. To assess Hypothesis 2, Spearman's correlation analysis examining the associations between brooding rumination, depression and PTSD symptoms were conducted in the overall sample, and for each separate group. To test Hypothesis 3, two moderation analyses (model 1) using PROCESS v3.5 [39] were conducted to determine whether cultural group moderated the associations between 1) brooding rumination and depression symptoms, and 2) brooding rumination and PTSD symptoms. Significance of results was indicated by 95% confidence intervals not including zero.

## Results

Table 1 presents participant characteristics and shows significant group differences for age, education, religion and time since trauma.

### Hypothesis 1

As shown in Table 1, Malaysian participants had significantly greater brooding rumination than Australian participants, $F(1, 228) = 5.77$, $p = .02$, $\eta_p^2 = .03$.

### Hypothesis 2

There was a statistically significant, moderate positive correlation between brooding rumination and depression, $r_s(251) = .42$, 95%CI [.29, .52], $p < .001$, and between PTSD and brooding rumination, $rs(226) = .51$, 95%CI [.40, .60], $p < .001$. When examining each cultural group separately, there was a statistically significant positive associations between brooding rumination and depression in the Malaysian, $r_s(142) = .50$, 95%CI [.36, .62], $p < .001$, and Australian groups, $r_s(107) = .31$, 95%CI [.10, .50], $p = .001$, and between brooding rumination and PTSD in the Australian, $rs(105) = .43$, 95%CI [.26, .57], $p < .001$ and Malaysian groups, $rs(119) = .59$, 95%CI [.45, .70], $p < .001$. The PTSD findings were similar when trauma type and time since trauma were included as covariates.

### Hypothesis 3

Table 2 presents the results of the moderation analyses. There was no evidence that cultural group moderated the associations between brooding rumination and depression, $F(1, 249) = 3.03$, $p = .08$, $R^2\Delta = .01$, or brooding rumination and PTSD symptoms, $F(1, 220) = 2.06$,

**Table 1. Participant characteristics.**

|  | Australian Group | Malaysian Group | Statistic |
|---|---|---|---|
| Age (years) | 31.53 (12.68) | 25.88 (7.64) | $t(165.2) = 4.06$** |
| Gender[a] | 21:87:1 | 32:108:4 | $\chi^2(2, N = 253) = 1.53$ |
| Education[b] | 27:21:39:20:2 | 12:10:94:20:8 | $\chi^2(4, N = 253) = 31.78$** |
| Religion[c] | 36:57:7:0:0:7:2 | 3:22:48:37:18:7:9 | $\chi^2(5, N = 253) = 127.46$** |
| HADS-D | 5.8 (3.76) | 5.65 (3.83) | $t(251) = .30$ |
| Index trauma[d,e] | 35:15:15:28:8:6 | 49:17:12:21:14:8 | $\chi^2(6,228) = 6.26$ |
| Time since trauma[d] | 10.18(12.38) | 5.87(5.75) | $t(146) = 3.30$** |
| PCL-5[d] | 22.96(17.68) | 22.69(18.45) | $t(226) = .12$ |
| RRS-B | 11.32 (3.69) | 11.54 (3.7) | $F(1,228) = 5.77$* |

*Note*. Data shown is the mean (SD) for all characteristics except gender, religion and education

* $p < .05$

** $p < .01$

HADS-D = Hospital Anxiety and Depression Scale (Depression subscale); PCL-5 = PTSD Checklist for DSM-5; RRS-B = Ruminative Response Scale (Brooding subscale)

[a] Male: female: prefer not to say

[b] Self-reported highest education level attained: secondary education: post-secondary education: undergraduate degree: postgraduate degree: other

[c] Self-reported religious background: no religion: Christianity: Islam or Muslim: Buddhism or Taoism: Hinduism: Other (including Sikhism and Agnosticism): did not reveal religion

[d] For the PTSD data only data from a subset of the overall sample was used.

[e] Index trauma category, displayed as accident, serious injury or illness: non-sexual assault or abuse: sexual assault: witnessing death: war or natural disaster: other.

**Table 2. Summary of moderation analyses.**

| Model | | Coefficient | | | t | p |
|---|---|---|---|---|---|---|
| | | B | SE | [95% CI] | | |
| Depression | Constant | 5.60 | 0.29 | [5.04–6.16] | 19.63 | < .001 |
| | RRS-B score | 0.53 | 0.08 | [.38-.68] | 6.83 | < .001 |
| | Cultural group | 0.24 | 0.43 | [-.62–1.09] | 0.54 | .59 |
| | Interaction | -0.21 | 0.12 | [-.44-.03] | -1.74 | .08 |
| PTSD | Constant | 23.20 | 5.68 | [11.82, 34.49] | 4.17 | < .001 |
| | RRS-B score | 3.77 | 0.97 | [1.84, 5.64] | 4.41 | < .001 |
| | Cultural group | 0.63 | 2.39 | [-3.89, 5.38] | 0.25 | .80 |
| | Interaction | -0.81 | 0.64 | [-2.07, 0.40] | -1.43 | .15 |

*Note*: PTSD = Posttraumatic Stress Disorder. RRS-B Ruminative Response Scale (Brooding subscale)

$p = .15$, $R^2\Delta < .01$. The PTSD findings were similar when trauma type and time since trauma were included as covariates.

## Discussion

This study investigated the moderating effect of cultural group (Australia vs Malaysia) on the relationships between brooding rumination and symptoms of depression and PTSD. In support of Hypothesis 1, we found that Malaysian participants had higher levels of brooding rumination compared to Australian participants. In support of Hypothesis 2, brooding rumination was significantly associated with both greater depression and PTSD symptom severity. Inconsistent with Hypothesis 3, we did not find a significant moderating effect of cultural group on the relationship between brooding rumination and either depression or PTSD symptom severity.

Our finding that Malaysians reported higher levels of brooding rumination compared to Australians aligns with previous cross-cultural non-clinical research; members of collectivistic cultures (e.g., Malaysia) tend to brood more than members of individualistic cultures (e.g., Australia) [18, 29, 40]. In relation to Hypothesis 2, we found that brooding rumination was positively associated with both depression and PTSD symptom severity in both Australian and Malaysian groups, which also aligns with previous literature indicating that brooding rumination is associated with increased psychopathology [9, 41].

However, there was no evidence of a moderating effect of cultural group on the relationship between brooding rumination and either symptoms of PTSD or depression. This observation does not align with the expected cross-cultural differences theorized and observed in non-clinical research which indicates brooding rumination is less associated with psychopathology for Asian individuals than Western individuals [16–18]. Our results suggest that these cultural differences observed in the non-clinical literature may not extend to clinical populations. It is possible that at higher levels of psychopathology, the strong negative effect of brooding rumination may override the cultural differences found in healthy populations. Such a notion is consistent with existing literature which implicates high levels of brooding rumination as a transdiagnostic maintenance factor across many disorders [41], which may play a role regardless of cultural background [23]. Thus, when considering psychopathology, brooding rumination may be similarly maladaptive.

If such findings are robust and replicable across other cultures, existing clinical models, which implicate brooding rumination in the maintenance of depression and PTSD (e.g., [7, 11]), may be applicable cross-culturally. Additionally, as existing evidence-based treatments for

depression and PTSD recommend brooding rumination as a key target [8, 42, 43], such a target may be suitable for Malaysian clients. However, future research is needed to further understand the content and form of brooding rumination in Malaysian clinical populations and the influence of culture on the role of brooding rumination in psychopathology.

The study had several limitations. First, we did not measure participants' cultural views and beliefs, and instead utilised nationality (Australian or Malaysian) and cultural heritage as an indicator of cultural orientation. While this approach was consistent with previous cross-cultural research [29, 44], future research could also examine cultural values at the individual level [45]. Second, we did not utilise a clinical sample. Third, the cross-sectional study design does not allow for causal inferences to be made. Finally, there were significantly more females than males in our study. Females tend to ruminate more on their depressive symptoms [46, 47]. Consequently, the mean levels of brooding rumination and depression symptoms could have been inflated by the imbalanced gender ratios of participants in the study. Despite these limitations, we found Malaysians had higher levels of brooding rumination than Australians, and that brooding rumination was positively associated with PTSD and depression symptom severity in both cultural groups.

## Acknowledgments

A special thanks to all the participants that entered the study.

## Author contributions

**Conceptualization:** James Haoxiang Li, Bryan Lee, Tamsyn Reyneke, Shamsul Haque, Siti Zainab Abdullah, Britney Kerr Wen Tan, Belinda Liddell, Laura Jobson.

**Data curation:** James Haoxiang Li, Bryan Lee, Tamsyn Reyneke, Shamsul Haque, Siti Zainab Abdullah, Britney Kerr Wen Tan, Laura Jobson.

**Formal analysis:** James Haoxiang Li, Bryan Lee, Tamsyn Reyneke, Siti Zainab Abdullah, Britney Kerr Wen Tan.

**Investigation:** James Haoxiang Li, Bryan Lee, Tamsyn Reyneke, Siti Zainab Abdullah, Britney Kerr Wen Tan.

**Methodology:** James Haoxiang Li, Bryan Lee, Tamsyn Reyneke, Siti Zainab Abdullah, Britney Kerr Wen Tan.

**Project administration:** James Haoxiang Li, Bryan Lee, Tamsyn Reyneke, Shamsul Haque, Siti Zainab Abdullah, Britney Kerr Wen Tan, Laura Jobson.

**Resources:** Shamsul Haque.

**Supervision:** Shamsul Haque, Belinda Liddell, Laura Jobson.

**Writing – original draft:** James Haoxiang Li, Bryan Lee, Tamsyn Reyneke, Siti Zainab Abdullah, Britney Kerr Wen Tan.

**Writing – review & editing:** James Haoxiang Li, Bryan Lee, Tamsyn Reyneke, Shamsul Haque, Siti Zainab Abdullah, Britney Kerr Wen Tan, Belinda Liddell, Laura Jobson.

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
