## [Decision Letter · Decision Letter 0]

12 Sep 2022

PONE-D-22-19386Does Culture Moderate the Relationships between Rumination and Symptoms of Posttraumatic Stress Disorder and Depression?PLOS ONE

Dear Dr. Bryan Lee,

Thank you for submitting your manuscript to PLOS ONE. After careful consideration, we feel that it has merit but does not fully meet PLOS ONE’s publication criteria as it currently stands. Therefore, we invite you to submit a revised version of the manuscript that addresses the points raised during the review process.

We look forward to receiving your revised manuscript.

Kind regards,

Rogis Baker, Ph.D

Academic Editor

PLOS ONE

Journal Requirements:

2. Please provide additional details regarding participant consent. In the Methods section, please ensure that you have specified (1) whether consent was informed and (2) what type you obtained (for instance, written or verbal). If your study included minors, state whether you obtained consent from parents or guardians. If the need for consent was waived by the ethics committee, please include this information. 

3. Please change "female” or "male" to "woman” or "man" as appropriate, when used as a noun (see for instance https://apastyle.apa.org/style-grammar-guidelines/bias-free-language/gender ). 

4. Peer review at PLOS ONE is not double-blinded (https://journals.plos.org/plosone/s/editorial-and-peer-review-process ). For this reason, authors should include in the revised manuscript all the information removed for blind review.

Reviewers' comments:

Reviewer's Responses to Questions

**Comments to the Author**

1. Is the manuscript technically sound, and do the data support the conclusions?

Reviewer #1: Yes

Reviewer #2: Yes

Reviewer #3: Partly

Reviewer #4: Yes

2. Has the statistical analysis been performed appropriately and rigorously?

Reviewer #1: Yes

Reviewer #2: I Don't Know

Reviewer #3: Yes

Reviewer #4: Yes

3. Have the authors made all data underlying the findings in their manuscript fully available?

Reviewer #1: Yes

Reviewer #2: Yes

Reviewer #3: Yes

Reviewer #4: Yes

4. Is the manuscript presented in an intelligible fashion and written in standard English?

Reviewer #1: Yes

Reviewer #2: Yes

Reviewer #3: Yes

Reviewer #4: Yes

5. Review Comments to the Author

Reviewer #1: The paper by Lee et al. aims to determine if culture modifies the relationship between rumination and symptoms of both PTSD and depression. They cover a pertinent topic using valid and strong statistical methods.

I only have minor comments regarding table 1 for it to be more reader friendly. Please consider the following recommendations.

1. Include the sample size of both groups at the top of the columns.

2. Divide the subcategories for gender, education, religion, and index trauma so that they have a separate row. If so, you could also consider including the (%) for each subcategory. - For example: Women 21 (19%).

3. For the variables that have a subset of the overall sample used include the n of the subset.

Reviewer #2: The authors have stated whether the cultural differences modify the relationships between brooding rumination and psychopathological symptoms among Malaysian and western driven Australian population and produced important findings.

Comments: The title doesn’t tell us the entire aim of the MS; I suggest modifying as per the objectives stated in the main document

Method:

The Rumination Response Scale (RRS) used in the current study is the one with only five items, while studies have been reporting 22 items to measure similar psychological symptoms, and not extensively described, this needs justification or to be stated under limitations

In addition, depressive symptoms were measured using HADS_D, while the preferred GHL-12 tool available, justify for using the short scale. Or else, indicate as a limitations

Reviewer #3: Comments

Clarifying the lack in scientific community regarding cultural influence in the associations between brooding rumination and symptoms of depression and PTSD, is a very important research to fill the existing gap in the research world. The author has raised a good topic but the manuscript does not follow the necessary findings. The introduction, methodology, result, discussion and conclusion section need to be scientifically written. Some major revisions are required before it could be considered for publication as follows:

I have doubt regarding the generalizability of your study to other world communities from different cultural background. The study may be failed when we think of other communities and how do you manage this? I mean do you think that cultural group did not moderate the relationships between brooding rumination and symptoms of depression and PTSD for different communities from different cultural background? It should be limitation for this study because you wrote that accumulating research suggests that the relationship between brooding rumination and psychopathology may vary between different cultural groups.

Authors are required to re-write the abstract. Authors must show how study participants were selected, how data were analyzed and the study good results in the abstract section to enhance the readers.

Please concrete the keywords and make them formally and academically. You have repeated “PTSD; post-traumatic stress disorder;”, so try to write/use only one of them.

Authors must check the manuscript carefully before submitting it to the journal. Because still manuscript having grammar mistake and spellings mistake. English language needs polishing throughout the manuscript. Have it edited by a professional.

To avoid any plagiarism and integrate the concepts from different researches of your interest, please give references at the end of each sentence for the paragraph “However, this evidence base is almost entirely derived from Western samples. Consequently, the dominant paradigm that suggests rumination is universally associated with increased psychopathology, is a notion based largely on Western research and cultural values. This is a key limitation given that (1) globally, many individuals with depression and PTSD do not come from Western cultural backgrounds, and (2) accumulating research suggests that the relationship between brooding rumination and psychopathology may vary between different cultural groups.”

You have described that brooding rumination has been found to be less associated with poor psychological outcomes, including anxiety, depression and life dissatisfaction, in Asian individuals compared to Western individuals. So, what you would do if your study might come with no association?

You have tried to address three hypotheses, but your title only focuses on the cultural influence. What about the other two hypotheses “brooding rumination would be positively associated with both depression and PTSD symptom severity” and “there would be greater brooding rumination reported by the Malaysian group than the Australian group?

I think it is better to say something in the method section as a “study area” about Australian and Malaysian, their culture, prevalence of brooding rumination, depression and PTSD symptom as characteristics of the community may be from other studies.

You have used convenience sampling method, and do you think your study is representative to the general population of Australian and Malaysian who have brooding rumination? If not, what is the significance of the study?

Do you think 253 study participant is enough to say a research for such large populated countries? It is better to increase your sample size which may affect your current result.

Try to clearly explain the following criteria in your document, why age between 18 and 65?

Please provide any reference for this “PTSD Checklist for DSM-5 (PCL-5) with Life Events Checklist for DSM-5 (LEC-5)” from previously conducted studies on this area.

If you have ethical approval obtained from BLINDED, please try to attach.

How did you assured that the data you collected have good quality? Are you confident for saying I had good research tool because of its co-design by Australian and Malaysian researchers?

It there any other additional information you collected from participants before the study begins to check whether they are real participants who fulfilled the criteria or not? helpful to control biases.

You have not described the result of the study in detail. First, prepare one table independently for socio-demographic characteristics. Try to describe the socio-demographic characteristics of the participant’s independently in one table and discuss within one or two paragraph about the result in the table. You have to describe the result of your current tables’ one and two in detail within at least each two paragraphs above the tables.

If you did not find a significant moderating effect of cultural group on the relationship between brooding rumination and either depression or PTSD symptom severity, what you would recommend in relation to other researches that reported culture influence in the association?

Our finding that Malaysians reported higher levels of brooding rumination compared to Australians aligns with previous cross-cultural non-clinical research; members of collectivistic cultures (e.g., Malaysia) tend to brood more than members of individualistic cultures (e.g., Australia). What is your justification for this finding or reason?

It is better to separate your discussion section into two separate sections “Discussion, and conclusion”. Discuss you conclusion with recommendations?

The discussion is too short, so it needs both detail internal and external discussions with clear justifications.

You have a lot of limitation, and I think it will have effect on the acceptability of you result for publication. So, try to reduce limitations and display only core limitations in your discussion section.

Please try to include abbreviation section and authors’ contribution section at the end.

Try to number and heading the titles and subtitles of your document starting from introduction.

Reviewer #4: The paper needs some minor revisions. I indicated by highlighting those issues to be addressed in the manuscript as well as through my review report. 

6. PLOS authors have the option to publish the peer review history of their article (what does this mean? ). If published, this will include your full peer review and any attached files.

Reviewer #1: No

Reviewer #2: No

Reviewer #3: No

Reviewer #4: Yes

---

## [Author Response · Author response to Decision Letter 0]

2 Nov 2022

Editor:

Apologies for the error. We have changed the file names to be consistent with the style requirements.

Please provide additional details regarding participant consent. In the Methods section, please ensure that you have specified (1) whether consent was informed and (2) what type you obtained (for instance, written or verbal). If your study included minors, state whether you obtained consent from parents or guardians. If the need for consent was waived by the ethics committee, please include this information.

Please change "female” or "male" to "woman” or "man" as appropriate, when used as a noun

We have changed “female” and “male” to “woman” or “man” as appropriate.

Peer review at PLOS ONE is not double-blinded. For this reason, authors should include in the revised manuscript all the information removed for blind review.

We have added the information removed for blind review “Ethical approval was obtained from the Monash University Human Research Ethics Committee (approval number MRHREC 27577).”

Please include your full ethics statement in the ‘Methods’ section of your manuscript file. In your statement, please include the full name of the IRB or ethics committee who approved or waived your study, as well as whether or not you obtained informed written or verbal consent. If consent was waived for your study, please include this information in your statement as well.

We have added the name of the ethics committee “Ethical approval was obtained from the Monash University Human Research Ethics Committee (approval number MRHREC 27577).”

We have added a statement of informed consent:” Participants first read an explanatory statement in Qualtrics and their decision to continue to the questionnaire was regarded as informed consent.”

Reviewer 1:

Include the sample size of both groups at the top of the columns.

We have added the sample size of both groups as suggested

Divide the subcategories for gender, education, religion, and index trauma so that they have a separate row. If so, you could also consider including the (%) for each subcategory. - For example: Women 21 (19%).

We have considered this recommendation and believe the table is more concise without the additional rows.

For the variables that have a subset of the overall sample used include the n of the subset.

We have added a note at the bottom of the table: “For the PTSD data only data from a subset of the overall sample was used (n = 228).” This information can also be found in the statistical analyses section of the report.

Reviewer 2:

The title doesn’t tell us the entire aim of the MS; I suggest modifying as per the objectives stated in the main document

The title has been amended

Method:

The Rumination Response Scale (RRS) used in the current study is the one with only five items, while studies have been reporting 22 items to measure similar psychological symptoms, and not extensively described, this needs justification or to be stated under limitations

In addition, depressive symptoms were measured using HADS_D, while the preferred GHL-12 tool available, justify for using the short scale. Or else, indicate as a limitations

We have added our justification for using the HADS-D: “Despite the availability of other depression measures (e.g., General Health Questionnaire-12), the HADS-D was selected on the basis of its strong validity and reliability, including in cross-cultural research [42]. Additionally, the use of a shorter scale aided in reducing participant fatigue.”

We have added our justification for using the five-item RRS-B: “Although rumination is typically measured using the 22-item Ruminative Response Scale, this study focused on the measurement of brooding rumination [45]. Hence, brooding rumination was measured using the brooding subscale of the RRS-SF (RRS-B) [45]. The utilisation of a shorter RRS scale also served to reduce participant fatigue.”

Reviewer 3:

I have doubt regarding the generalizability of your study to other world communities from different cultural background. The study may be failed when we think of other communities and how do you manage this? I mean do you think that cultural group did not moderate the relationships between brooding rumination and symptoms of depression and PTSD for different communities from different cultural background? It should be limitation for this study because you wrote that accumulating research suggests that the relationship between brooding rumination and psychopathology may vary between different cultural groups.

We have removed the paragraph relating to generalizability and instead added a paragraph that highlights these points.

“Despite our moderation findings not aligning with previous research, it is important to note that this study just focused on a sample of specific Malaysian and Australian community members. Thus, our findings cannot be generalized to other communities and to those from different cultural backgrounds. Moreover, we used convenience sampling method and thus our findings may not be representative to the general population of Australia and Malaysia, particularly given the population size of these countries and diversity within these countries. Additionally, the findings may differ in clinical samples. Nevertheless, this study is important as this area is exceptionally under- researched and this study highlights the importance of continuing to investigate the relationships between brooding rumination and depression and PTSD in different cultural contexts. This is a complex area, as culture and emotion regulation are complex constructs[55,56]. Additionally, research needs to examine the applicability of PTSD and depression models and their accounts of rumination in other cultural contexts and whether cultural variables influence the processes posited in these models.” (Page 13)

Authors are required to re-write the abstract. Authors must show how study participants were selected, how data were analyzed and the study good results in the abstract section to enhance the readers.

We have added how participants were selected: “European Australians (n= 109) and Malaysians of varying Asian heritages (n= 144) from the community were recruited through social media platforms (Facebook & Gumtree).”

We have added how data was analysed: “Data was analysed using a one-way analysis of covariance to compare levels of BR between the Malaysian and Australian groups; Spearman’s correlation analyses examined the associations between BR, depression and PTSD symptoms; and Two moderation analyses were conducted to determined whether cultural group moderated the associations between BR and both depression and PTSD symptoms”.

We have added further details about the results to the abstract.

Please concrete the keywords and make them formally and academically. You have repeated “PTSD; post-traumatic stress disorder;”, so try to write/use only one of them.

We have removed post-traumatic stress disorder as a keyword

Authors must check the manuscript carefully before submitting it to the journal. Because still manuscript having grammar mistake and spellings mistake. English language needs polishing throughout the manuscript. Have it edited by a professional.

Apologies for the errors. We have now carefully checked the document.

To avoid any plagiarism and integrate the concepts from different researches of your interest, please give references at the end of each sentence for the paragraph “However, this evidence base is almost entirely derived from Western samples. Consequently, the dominant paradigm that suggests rumination is universally associated with increased psychopathology, is a notion based largely on Western research and cultural values. This is a key limitation given that (1) globally, many individuals with depression and PTSD do not come from Western cultural backgrounds, and (2) accumulating research suggests that the relationship between brooding rumination and psychopathology may vary between different cultural groups.”

References have been inserted.

You have described that brooding rumination has been found to be less associated with poor psychological outcomes, including anxiety, depression and life dissatisfaction, in Asian individuals compared to Western individuals. So, what you would do if your study might come with no association?

We have added greater details here: “This observation does not align with the expected cross-cultural differences theorized and observed in non-clinical research which indicates brooding rumination is less associated with psychopathology for Asian individuals than Western individuals[19-21]. We cannot be certain why our results do not align with these previous findings. It is possible that cultural differences observed in the non-clinical literature do not extend to clinical populations. Given the sophisticated theoretical accounts outlined above positing why rumination maintains symptoms of depression and PTSD, it is plausible that such negative influences on cognitive and affective processing associated with these two disorders override cultural effects.” (page 12-13)

You have tried to address three hypotheses, but your title only focuses on the cultural influence. What about the other two hypotheses “brooding rumination would be positively associated with both depression and PTSD symptom severity” and “there would be greater brooding rumination reported by the Malaysian group than the Australian group?

Thank you for these suggestions: We have changed the title to ‘Cultural Differences in Brooding Rumination in Depression and Posttraumatic Stress Disorder’

I think it is better to say something in the method section as a “study area” about Australian and Malaysian, their culture, prevalence of brooding rumination, depression and PTSD symptom as characteristics of the community may be from other studies.

As noted above (point 12) this information has been now added to the Introduction. We have also included more information in the Study Design Section.

You have used convenience sampling method, and do you think your study is representative to the general population of Australian and Malaysian who have brooding rumination? If not, what is the significance of the study?

We have removed the paragraph relating to generalizability and instead added a paragraph that highlights these points.

“Despite our moderation findings not aligning with previous research, it is important to note that this study just focused on a sample of specific Malaysian and Australian community members. Thus, our findings cannot be generalized to other communities and to those from different cultural backgrounds. Moreover, we used convenience sampling method and thus our findings may not be representative to the general population of Australia and Malaysia, particularly given the population size of these countries and diversity within these countries. Additionally, the findings may differ in clinical samples. Nevertheless, this study is important as this area is exceptionally under- researched and this study highlights the importance of continuing to investigate the relationships between brooding rumination and depression and PTSD in different cultural contexts. This is a complex area, as culture and emotion regulation are complex constructs[55,56]. Additionally, research needs to examine the applicability of PTSD and depression models and their accounts of rumination in other cultural contexts and whether cultural variables influence the processes posited in these models.” (Page 13)

Do you think 253 study participant is enough to say a research for such large populated countries? It is better to increase your sample size which may affect your current result.

For an initial study this sample size is appropriate and was justified by an a-priori power calculation. Future studies should now include larger sample sizes to examine whether these findings are replicable and to further explore this area – an area in much need of research.

Try to clearly explain the following criteria in your document, why age between 18 and 65?

We have changed eligibility to inclusion criteria.

We have justified the age range selection: “The age range was selected because this study focused on adults; previous cross-cultural clinical studies have similarly focused on this age range [14,57]; and research indicates that rumination changes over the lifespan, with older adults being significantly less likely to engage in brooding rumination than their younger counterparts [59].” (Page 7).

Please provide any reference for this “PTSD Checklist for DSM-5 (PCL-5) with Life Events Checklist for DSM-5 (LEC-5)” from previously conducted studies on this area.

We have added “The PCL-5 with LEC-5[42] is the gold-standard self-report scale used to assessed trauma exposure and PTSD symptom severity” (page 7)

We have also added: The PCL-5 has demonstrated good psychometric properties, including in cross-cultural research[43] and cultural rumination research [44]. (Page 7)

If you have ethical approval obtained from BLINDED, please try to attach.

We have added the ethical approval number: approval number MRHREC 27577 (page 10). We have also included the approval letter

How did you assured that the data you collected have good quality? Are you confident for saying I had good research tool because of its co-design by Australian and Malaysian researchers?

We are confident that the data is good quality as:

- We had several quality checks (outlined below)

- The research inspected all data and open-end responses to ensure responding was valid

- We had researchers from both countries providing input into the design, measures and interpretation of findings

- We used measures with strong psychometric properties routinely used in the mental health literature, including in cross-cultural literature, and all measures had good internal consistency in our study

- All measures were translated using gold-standard procedures

It there any other additional information you collected from participants before the study begins to check whether they are real participants who fulfilled the criteria or not? helpful to control biases.

We have added: “To ensure quality data, the researchers undertook several data checks prior to the analyses; fraud detection in Qualtrics was used to flag potential bot responses; flagged responses, open ended text responses and demographic information were visually inspected independently by three researchers; and the CRS was used to identify random responders and bots.” (Page 8)

You have not described the result of the study in detail. First, prepare one table independently for socio-demographic characteristics. Try to describe the socio-demographic characteristics of the participant’s independently in one table and discuss within one or two paragraph about the result in the table. You have to describe the result of your current tables’ one and two in detail within at least each two paragraphs above the tables.

We have provided further details of the results. We have added greater descriptions of Table 1 and socio-demographics of the sample. “Table 1 presents group characteristics. As shown in Table 1, no between-group differences were found for gender, depression symptoms, or PTSD symptoms. However, significant between-group differences were found for age, such that Australian participants were significantly older than Malaysian participants. Significant group differences were also found for education, religion and time since trauma. Thus, we also conducted the below hypothesis-related analyses including age, education, religion, and time since trauma as covariates. In each instance the pattern of results remained consistent to that reported.”(pages 9-10).

We don’t believe two tables are necessary as the second table will only have the PCL-5 and RRS-B in it so to make it more concise we have included all details Table 1. We have more clearly referred to the Tables though in the text.

If you did not find a significant moderating effect of cultural group on the relationship between brooding rumination and either depression or PTSD symptom severity, what you would recommend in relation to other researches that reported culture influence in the association?

As noted above, in the Discussion we have now highlighted the need for further research in this area.

Our finding that Malaysians reported higher levels of brooding rumination compared to Australians aligns with previous cross-cultural non-clinical research; members of collectivistic cultures (e.g., Malaysia) tend to brood more than members of individualistic cultures (e.g., Australia). What is your justification for this finding or reason?

We have added: “Specifically, past research has demonstrated that; members of collectivistic cultures (e.g., Malaysia) tend to brood more than members of individualistic cultures (e.g., Australia) as brooding rumination can function in order to maintain interpersonal harmony and prevent future disruptions to the group[16,21,40]. Those from collectivistic cultures have also been proposed to adopt a more self-distanced approach to rumination[40]. Additionally, as suggested by other previous researchers have accounted for this cultural difference by noting that literature, Asians may dwell on past events more frequently in order to engage in self-criticism, which may serve to help improve oneself or support the group and interdependence [51,52]. Thus, this first finding supports accumulating research demonstrating that those with an Asian cultural background ruminate more than those from a Western European background[18,40].” (Page 11)

It is better to separate your discussion section into two separate sections “Discussion, and conclusion”. Discuss you conclusion with recommendations?

We have added implications and conclusions subheading

The discussion is too short, so it needs both detail internal and external discussions with clear justifications.

We have re-worked the Discussion so there is greater depth and discussion.

You have a lot of limitation, and I think it will have effect on the acceptability of you result for publication. So, try to reduce limitations and display only core limitations in your discussion section.

We have modified the limitations section, so it just now focuses on core limitations.

Please try to include abbreviation section and authors’ contribution section at the end.

We have added a contribution section

Try to number and heading the titles and subtitles of your document starting from introduction.

We have added numbers for the headings and subheadings.

Reviewer 4:

This title likely deserves a binary response “Yes/No”. Then, what will happen after “Yes/No” response? It seems the last answer to the research question. I strongly recommend that the authors should avoid interrogation in writing the title of the paper. Instead, they can rewrite it affirmative sentence as follows: “The Role of Culture in Moderating the Relationships between Rumination and Symptoms of Posttraumatic Stress Disorder and Depression” Or, using what is provided as a short title would be sufficient: cultural differences in rumination in depression and PTSD (in expanded form).

Thank you for these suggestions: We have changed the title to ‘Cultural Differences in Brooding Rumination in Depression and Posttraumatic Stress Disorder’

The first sentence is like your research finding. In practice however, you should start with brief introduction to the problem (may be in a sentence). Is it based on your research result or just a kind of problem statement. In practice, you need to start by stating the problem in brief followed by aim/objective of the paper, methodology, main findings and conclusion. For that matter, the relationship between brooding rumination & symptoms of depression & PTSD may not be always positive.

We have added: “Brooding rumination is a transdiagnostic risk factor for both depression and posttraumatic stress disorder (PTSD).”

We have reworded the second sentence to reflect that the relationship may not always be positive: “Although brooding rumination has typically been found to be positively associated with depression and PTSD symptoms, emerging non-clinical cross-cultural research indicates that culture may influence these associations.”

We have added how participants were selected: “European Australians (n= 109) and Malaysians of varying Asian heritages (n= 144) from the community were recruited through social media platforms (Facebook & Gumtree).”

We have added how data was analysed: “Data was analysed using a one-way analysis of covariance to compare levels of BR between the Malaysian and Australian groups; Spearman’s correlation analyses examined the associations between BR, depression and PTSD symptoms; and Two moderation analyses were conducted to determined whether cultural group moderated the associations between BR and both depression and PTSD symptoms”.

We have added further details about the results to the abstract.

In the Keywords, avoid one of the words (PTDS or post-traumatic stress disorder) and use one of them.

We have removed post-traumatic stress disorder as a keyword

This leads the reader to understand that the only factor causing differences in the RelationshipsbetweenRuminationandSymptomsofPosttraumaticStressDisorderandDepression. However, many more factors are responsible for such differences. So, the finding you arrived at and the conclusion drawn may not be reliable unless you could statistically capture those extraneous variables/factor that may also be responsible for such differences. So, how can you harmonize this gap between your ‘culture only variable” and “the impacts of other non-cultural variables”?

This is an important point, so we have added to the end of Paragraph 1: “This study focuses on one such factor, culture.” (page 3).

The overall idea of this paragraph is that cultural differences in brooding rumination has been conducted but not in the study area because such studies are concentrated in western countries but not in Australia. Well, for one thing the authors must exhaustively reviewed literature to be sure of this. Second, the idea that previous literature established the theory that there is such cultural difference in brooding rumination can also apply to many other cultures. As a result, the authors may be asked what the need to conduct the research in Australia is. They need to justify this very clearly.

We have updated this section: “Additionally, cross-cultural non-clinical research in this area is somewhat limited by its sampling of mostly American and Chinese populations. Cross-cultural researchers have highlighted the importance of exploring the relationships between psychological processes and psychological adjustment in varying Western individualistic and Asian collectivistic cultures [27]. In the current study, we aimed to extend this area of cross-cultural research by focusing on Australia (as a Western cultural context [28]) and Malaysia (as an Asian cultural context[25]). Australia is considered an individualistic society [28,29], while Malaysia – a multi-cultural society primarily populated by individuals of Malay, Chinese and Indian heritage – is considered a collectivistic society[30,31]. Importantly, both countries have high lifetime prevalence rates of depression (Australia 10%; Malaysia 10.3%) and PTSD (Australia 12%; Malaysia 8.5%) [32,33], which are associated with significant widespread social and economic burden (e.g., [34-36]). Given the central role rumination has in PTSD and depression, it is crucial to explore the relationships between culture, brooding rumination and symptoms of depression and PTSD in Australian and Malaysian cultural contexts.” (Page 5)

This is good but your topic is only on Yes/No type. However, here your hypotheses themselves talk much more than the binary responses that your topic informs the reader. So, rewrite your topic in line with these hypotheses.

This have been resolved after title amendment

What is the specific research design for this study? The authors should elaborate a bit about the basic research design immediately after this section. In my opinion, your research design is ‘cross-sectional survey”. Therefore, you would have explained how and why you employed the design before providing your sampling method.

We have added: “We utilised a cross-sectional, cross-country design to examine our hypotheses in Malaysia and Australia. The data was collected from March-May 2021. The target population included Malaysians and Australians aged 18-65 years and, following the approach of other recent similar studies[37], participants were reached using convenience and snowballing sampling methods. Participants completed an online survey (developed on Qualtrics) in English or Malay. Following the gold standard approach[38], English measures were translated into Malay and then back-translated into English. The study was promoted using social media (Gumtree and Facebook adverts) and those interested contacted the researchers and the online survey link was provided. The study was co-designed by researchers in Australia and Malaysia to ensure cultural appropriateness of design and interpretation of findings” (Page 6).

Was Convenience sampling or snowball sampling appropriate? This because the method by which the researchers accessed the respondents was via internet system (Facebook & Gumtree). Even, the method you used seems different from snowballing and convenience sampling. I suggest strongly to re-describe this method appropriately. Moreover, General community sample in the same paragraph is not clear as you cannot find it in a scientific circle. It seems your creation, otherwise try to mention some literatures that use similar phrase. This type of sampling is not known by the scientific community. Instead, elaborate the sampling technique you employed for the study…. In case how you employed a convenience sampling procedure although whether you employed convenience sampling method or not is still unclear.

Apologies this was not clearer. We have removed the term general community and have further described our sampling based on previous literature: “The target population included Malaysians and Australians aged 18-65 years and, following the approach of other recent similar studies[37], participants were reached using convenience and snowballing sampling methods. Participants completed an online survey (developed on Qualtrics) in English or Malay. Following the gold standard approach[38], English measures were translated into Malay and then back-translated into English. The study was promoted using social media (Gumtree and Facebook adverts) and those interested contacted the researchers and the online survey link was provided.” (Page 6)

I think the phrase ‘eligibility criteria’ is to mean inclusion and exclusion criteria. It is okay; but you have no reason to limit the age range of the respondents. In case you thought that you selected those who can read and fill out the questionnaire, at present children who completed their primary education can do that. Therefore, you should justify for limiting the age range you mentioned here.

We have changed eligibility to inclusion criteria.

We have justified the age range selection: “The age range was selected because this study focused on adults; previous cross-cultural clinical studies have similarly focused on this age range [14,57]; and research indicates that rumination changes over the lifespan, with older adults being significantly less likely to engage in brooding rumination than their younger counterparts [59].” (Page 7).

The serious problem lies here. This section is too shallow to draw the conclusion. The authors did not discussed their findings with sufficient number of related previous literature. They mainly focused on how the findings against each hypothesis agrees or disagrees each other. I recommend the authors should discuss further their findings in relation to previous literature in depth. The discussion is too short, so it needs both detail internal and external discussions with clear justifications.

We have re-worked the Discussion so there is greater depth and discussion.

---

## [Editor Report · Decision Letter 1]

15 Nov 2022

Cultural differences in brooding rumination in depression and posttraumatic stress disorder

PONE-D-22-19386R1

Dear Dr. Bryan Lee,

We’re pleased to inform you that your manuscript has been judged scientifically suitable for publication and will be formally accepted for publication once it meets all outstanding technical requirements.

An invoice for payment will follow shortly after the formal acceptance. To ensure an efficient process, please log into Editorial Manager at http://www.editorialmanager.com/pone/ , click the 'Update My Information' link at the top of the page, and double check that your user information is up-to-date. If you have any billing related questions, please contact our Author Billing department directly at authorbilling@plos.org.

Kind regards,

Rogis Baker, Ph.D

Academic Editor

PLOS ONE
---

## [Editor Report · Acceptance letter]

17 Nov 2022

PONE-D-22-19386R1

Does Culture Moderate the Relationships between Rumination and Symptoms of Posttraumatic Stress Disorder and Depression?

Dear Dr. Lee:

I'm pleased to inform you that your manuscript has been deemed suitable for publication in PLOS ONE. Congratulations! Your manuscript is now with our production department.

Kind regards,

on behalf of

Dr. Rogis Baker

Academic Editor

PLOS ONE